# GRAPH CLASSIFICATION WITH GEOMETRIC SCATTERING

## ABSTRACT

One of the most notable contributions of deep learning is the application of convolutional neural networks (ConvNets) to structured signal classification, and in particular image classification. Beyond their impressive performances in supervised learning, the structure of such networks inspired the development of deep filter banks referred to as scattering transforms. These transforms apply a cascade of wavelet transforms and complex modulus operators to extract features that are invariant to group operations and stable to deformations. Furthermore, ConvNets inspired recent advances in geometric deep learning, which aim to generalize these networks to graph data by applying notions from graph signal processing to learn deep graph filter cascades. We further advance these lines of research by proposing a geometric scattering transform using graph wavelets defined in terms of random walks on the graph. We demonstrate the utility of features extracted with this *designed* deep filter bank in graph classification of biochemistry and social network data (incl. state of the art results in the latter case), and in data exploration, where they enable inference of EC exchange preferences in enzyme evolution.

## 1 INTRODUCTION

Over the past decade, numerous examples have established that deep neural networks (i.e., cascades of linear operations and simple nonlinearities) typically outperform traditional "shallow" models in various modern machine learning applications, especially given the increasing Big Data availability nowadays. Perhaps the most well known example of the advantages of deep networks is in computer vision, where the utilization of 2D convolutions enable network designs that learn cascades of convolutional filters, which have several advantages over fully connected network architectures, both computationally and conceptually. Indeed, in terms of supervised learning, convolutional neural networks (ConvNets) hold the current state of the art in image classification, and have become the standard machine learning approach towards processing big structured-signal data, including audio and video processing. See, e.g., Goodfellow et al. (2016, Chapter 9) for a detailed discussion.

Beyond their performances when applied to specific tasks, pretrained ConvNet layers have been explored as image feature extractors by freezing the first few pretrained convolutional layers and then retraining only the last few layers for specific datasets or applications (e.g., Yosinski et al., 2014; Oquab et al., 2014). Such transfer learning approaches provide evidence that suitably constructed deep filter banks should be able to extract task-agnostic semantic information from structured data, and in some sense mimic the operation of human visual and auditory cortices, thus supporting the neural terminology in deep learning. An alternative approach towards such universal feature extraction was presented in Mallat (2012), where a deep filter bank, known as the scattering transform, is *designed*, rather than trained, based on predetermined families of distruptive patterns that should be eliminated to extract informative representations. The scattering transform is constructed as a cascade of linear wavelet transforms and nonlinear complex modulus operations that provides features with guaranteed invariance to a predetermined Lie group of operations such as rotations, translations, or scaling. Further, it also provides Lipschitz stability to small diffeomorphisms of the inputted signal. Scattering features have been shown to be effective in several audio (e.g., Bruna & Mallat, 2013a; Andén & Mallat, 2014; Lostanlen & Mallat, 2015) and image (e.g., Bruna & Mallat, 2013b; Sifre & Mallat, 2014; Oyallon & Mallat, 2015) processing applications, and their advantages over learned features are especially relevant in applications with relatively low data availability, such as quantum chemistry (e.g., Hirn et al., 2017; Eickenberg et al., 2017; 2018).

Following the recent interest in geometric deep learning approaches for processing graph-structured data (see, for example, Bronstein et al. (2017) and references therein), we present here a generalization of the scattering transform from Euclidean domains to graphs. Similar to the Euclidean case, our construction is based on a cascade of bandpass filters, defined in this case using graph signal processing (Shuman et al., 2013) notions, and complex moduli, which in this case take the form of absolute values (see Sec. 3). While several choices of filter banks could generally be used with the proposed cascade, we focus here on graph wavelet filters defined by lazy random walks (see Sec. 2). These wavelet filters are also closely related to diffusion geometry and related notions of geometric harmonic analysis, e.g. the diffusion maps algorithm of Coifman & Lafon (2006) and the associated diffusion wavelets of Coifman & Maggioni (2006). Therefore, we call the constructed cascade *geometric scattering*, which also follows the same terminology from geometric deep learning.

We note that similar attempts at generalizing the scattering transform to graphs have been presented in Chen et al. (2014) as well as Zou & Lerman (2018) and Gama et al. (2018). The latter two works are most closely related to the present paper. In them, the authors focus on theoretical properties of the proposed graph scattering transforms, and show that such transforms are invariant to graph isomorphism. The geometric scattering transform that we define here also possesses the same invariance property, and we expect similar stability properties to hold for the proposed construction as well. However, in this paper we focus mainly on the practical applicability of geometric scattering transforms for graph-structured data analysis, with particular emphasis on the task of graph classification, which has received much attention recently in geometric deep learning (see Sec. 4)

In supervised graph classification problems one is given a training database of graph/label pairs $\{(G_i, y_i)\}_{i=1}^N \subset \mathcal{G} \times \mathcal{Y}$ sampled from a set of potential graphs $\mathcal{G}$ and potential labels $\mathcal{Y}$. The goal is to use the training data to learn a model $f : \mathcal{G} \to \mathcal{Y}$ that associates to any graph $G \in \mathcal{G}$ a label $y = f(G) \in \mathcal{Y}$. These types of databases arise in biochemistry, in which the graphs may be molecules and the labels some property of the molecule (e.g., its toxicity), as well as in various types of social network databases. Until recently, most approaches were kernel based methods, in which the model $f$ was selected from the reproducing kernel Hilbert space generated by a kernel that measures the similarity between two graphs; one of the most successful examples of this approach is the Weisfeiler-Lehman graph kernel of Shervashidze et al. (2011). Numerous feed forward deep learning algorithms, though, have appeared over the last few years. In many of these algorithms, task based (i.e., dependent upon the labels $\mathcal{Y}$) graph filters are learned from the training data as part of the larger network architecture. These filters act on a characteristic signal $\mathbf{x}_G$ that is defined on the vertices of any graph $G$, e.g., $\mathbf{x}_G$ may be a vector of degrees of each vertex (we remark there are also edge based algorithms, such as Gilmer et al. (2017) and references within, but these have largely been developed for and tested on databases not considered in Sec. 4). Here, we propose an alternative to these methods in the form of a geometric scattering classifier (GSC) that leverages graph-dependent (but not label dependent) scattering transforms to map each graph $G$ to the scattering features extracted from $\mathbf{x}_G$. Furthermore, inspired by transfer learning approaches such as Oquab et al. (2014), we consider treatment of our scattering cascade as frozen layers on $\mathbf{x}_G$, either followed by fully connected classification layers (see Fig. 2), or fed into other classifiers such as SVM or logistic regression. We note that while the formulation in Sec. 3 is phrased for a single signal $\mathbf{x}_G$, it naturally extends to multiple signals by concatenating their scattering features.

In Sec. 4.1 we evaluate the quality of the scattering features and resulting classification by comparing it to numerous graph kernel and deep learning methods over 13 datasets (7 biochemistry ones and 6 social network ones) commonly studied in related literature. In terms of classification accuracy on individual datasets, we show that the proposed approach obtains state of the art results on two datasets and performs competitively on the rest, despite only learning a classifier that come after the geometric scattering transform. Furthermore, while other methods may excel on specific datasets, when considering average accuracy: within social network data, our proposed GSC outperforms all other methods; in biochemistry or over all datasets, it outperforms nearly all feed forward neural network approaches, and is competitive with state of the art results of graph kernels (Kriege et al., 2016) and graph recurrent neural networks (Taheri et al., 2018). We regard this result as crucial in establishing the universality of graph features extracted by geometric scattering, as they provide an effective task-independent representation of analyzed graphs. Finally, to establish their unsupervised qualities, in Sec. 4.2 we use geometric scattering features extracted from enzyme data (Borgwardt et al., 2005a) to infer emergent patterns of enzyme commission (EC) exchange preferences in enzyme evolution, validated with established knowledge from Cuesta et al. (2015).

## 2 GRAPH RANDOM WALKS AND GRAPH WAVELETS

We define graph wavelets as the difference between lazy random walks that have propagated at different time scales, which mimics classical wavelet constructions found in Meyer (1993) as well as more recent constructions found in Coifman & Maggioni (2006). The underpinnings for this construction arise out of graph signal processing, and in particular the properties of the graph Laplacian.

Let $G = (V, E, W)$ be a weighted graph, consisting of $n$ vertices $V = \{v_1, \ldots, v_n\}$, edges $E \subseteq \{(v_\ell, v_m) : 1 \leq \ell, m \leq n\}$, and weights $W = \{w(v_\ell, v_m) > 0 : (v_\ell, v_m) \in E\}$. Note that unweighted graphs are considered as a special case, by setting $w(v_\ell, v_m) = 1$ for each $(v_\ell, v_m) \in E$. Define the $n \times n$ (weighted) adjacency matrix $\mathbf{A}_G = \mathbf{A}$ of $G$ by $\mathbf{A}(v_\ell, v_m) = w(v_\ell, v_m)$ if $(v_\ell, v_m) \in E$ and zero otherwise, where we use the notation $\mathbf{A}(v_\ell, v_m)$ to denote the $(\ell, m)$ entry of the matrix $\mathbf{A}$ so as to emphasize the correspondence with the vertices in the graph and to reserve sub-indices for enumerating objects. Define the (weighted) degree of vertex $v_\ell$ as $\deg(v_\ell) = \sum_m \mathbf{A}(v_\ell, v_m)$ and the corresponding diagonal $n \times n$ degree matrix $\mathbf{D}$ given by $\mathbf{D}(v_\ell, v_\ell) = \deg(v_\ell)$, $\mathbf{D}(v_\ell, v_m) = 0$, $\ell \neq m$. Finally, the $n \times n$ graph Laplacian matrix $\mathbf{L}_G = \mathbf{L}$ on $G$ is defined as $\mathbf{L} = \mathbf{D} - \mathbf{A}$.

The graph Laplacian is a symmetric, real valued positive semi-definite matrix, and thus has $n$ non-negative eigenvalues. Furthermore, if we set $\mathbf{0} = (0, \ldots, 0)^T$ to to be the $n \times 1$ vector of all zeroes, and $\mathbf{1} = (1, \ldots, 1)^T$ to be the analogous vector of all ones, then it is easy to see that $\mathbf{L}\mathbf{1} = \mathbf{0}$. Therefore $0$ is an eigenvalue of $\mathbf{L}$ and we write the $n$ eigenvalues of $\mathbf{L}$ as $0 = \lambda_0 \leq \lambda_1 \leq \cdots \leq \lambda_{n-1}$ with corresponding $n \times 1$ orthonormal eigenvectors $\mathbf{1}/\sqrt{n} = \boldsymbol{\varphi}_0, \boldsymbol{\varphi}_1, \ldots, \boldsymbol{\varphi}_{n-1}$. If the graph $G$ is connected, then $\lambda_1 > 0$. In order to simplify the following discussion we assume that this is the case, although the discussion below can be amended to include disconnected graphs as well.

Since $\boldsymbol{\varphi}_0$ is constant and every other eigenvector is orthogonal to $\boldsymbol{\varphi}_0$, it is natural to view the eigenvectors $\boldsymbol{\varphi}_k$ as the Fourier modes of the graph $G$, with a frequency magnitude $\sqrt{\lambda_k}$. Let $\mathbf{x} : V \to \mathbb{R}$ be a signal defined on the vertices of the graph $G$, which we will consider as an $n \times 1$ vector with entries $\mathbf{x}(v_\ell)$. It follows that the Fourier transform of $\mathbf{x}$ can be defined as $\widehat{\mathbf{x}}(k) = \mathbf{x} \cdot \boldsymbol{\varphi}_k$, where $\mathbf{x} \cdot \mathbf{y}$ is the standard dot product. This analogy is one of the foundations of graph signal processing and indeed we could use this correspondence to define wavelet operators on the graph $G$, as in Hammond et al. (2011). Rather than follow this path, though, we instead take a related path similar to Coifman & Maggioni (2006); Gama et al. (2018) by defining the graph wavelet operators in terms of random walks defined on $G$, which will avoid diagonalizing $\mathbf{L}$ and will allow us to control the "spatial" graph support of the filters directly.

Define the $n \times n$ transition matrix of a lazy random random walk as $\mathbf{P} = \frac{1}{2}\left(\mathbf{D}^{-1}\mathbf{A} + \mathbf{I}\right)$. Note that the row sums of $\mathbf{P}$ are all one and thus the entry $\mathbf{P}(v_\ell, v_m)$ corresponds to the transition probability of walking from vertex $v_\ell$ to $v_m$ in one step. Powers of $\mathbf{P}$ run the random walk forward, so that in particular $\mathbf{P}^t(v_\ell, v_m)$ is the transition probability of walking from $v_\ell$ to $v_m$ in exactly $t$ steps. We will use $\mathbf{P}$ as a left multiplier, in which case $\mathbf{P}$ acts a diffusion operator. To understand this idea more precisely, first note that a simple calculation shows that $\mathbf{P}\mathbf{1} = \mathbf{1}$ and furthermore if the graph $G$ is connected, every other eigenvalue of $\mathbf{P}$ is contained in $[0, 1)$. Note in particular that $\mathbf{L}$ and $\mathbf{P}$ share the eigenvector $\mathbf{1}$. It follows that $\mathbf{P}^t\mathbf{x}$ responds most significantly to the zero frequency $\widehat{\mathbf{x}}(0)$ of $\mathbf{x}$ while depressing the non-zero frequencies of $\mathbf{x}$ (where the frequency modes are defined in terms of the graph Laplacian $\mathbf{L}$, as described above). On the spatial side, the value $\mathbf{P}^t\mathbf{x}(v_\ell)$ is the weighted average of $\mathbf{x}(v_\ell)$ with all values $\mathbf{x}(v_m)$ such that $v_m$ is within $t$ steps of $v_\ell$ in the graph $G$.

High frequency responses of $\mathbf{x}$ can be recovered in multiple different fashions, but we utilize multiscale wavelet transforms that group the non-zero frequencies of $G$ into approximately dyadic bands. As shown in Mallat (2012, Lemma 2.12), wavelet transforms are provably stable operators in the Euclidean domain, and the proof of Zou & Lerman (2018, Theorem 5.1) indicates that similar results on graphs may be possible. Furthermore, the multiscale nature of wavelet transforms will allow the resulting geometric scattering transform (Sec. 3) to traverse the entire graph $G$ in one layer, which is valuable for obtaining global descriptions of $G$. Following Coifman & Maggioni (2006), define the $n \times n$ diffusion wavelet matrix at the scale $2^j$ as

$$\mathbf{\Psi}_j = \mathbf{P}^{2^{j-1}} - \mathbf{P}^{2^j} = \mathbf{P}^{2^{j-1}}(\mathbf{I} - \mathbf{P}^{2^{j-1}}) \tag{1}$$

Since $\mathbf{P}^t\mathbf{1} = \mathbf{1}$ for every $t$, we see that $\mathbf{\Psi}_j\mathbf{1} = \mathbf{0}$ for each $j \geq 1$. Thus each $\mathbf{\Psi}_j\mathbf{x}$ partially recovers $\widehat{\mathbf{x}}(k)$ for $k \geq 1$. The value $\mathbf{\Psi}_j\mathbf{x}(v_\ell)$ aggregates the signal information $\mathbf{x}(v_m)$ from the vertices $v_m$

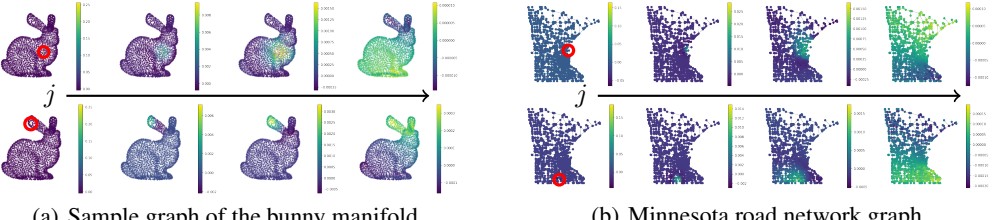

(a) Sample graph of the bunny manifold       (b) Minnesota road network graph

Figure 1: Diffusion wavelets $\boldsymbol{\Psi}_j$ for increasing scale $j$ left to right, applied to diracs centered at two different locations (marked by red circles) in two different graphs. Vertex colors indicate wavelet values (corresponding to colorbars for each plot), ranging from yellow/green indicating positive values to blue indicating negative values. Both graphs are freely available from PyGSP (2018).

that are within $2^j$ steps of $v_\ell$, but does not average the information like the operator $\mathbf{P}^{2^j}$. Instead, it responds to sharp transitions or oscillations of the signal $\mathbf{x}$ within the neighborhood of $v_\ell$ with radius $2^j$ (in terms of the graph path distance). Generally, the smaller $j$ the higher the frequencies $\boldsymbol{\Psi}_j\mathbf{x}$ recovers in $\mathbf{x}$. These high frequency wavelet coefficients up to the scale $2^J$ are denoted by:

$$\boldsymbol{\Psi}^{(J)}\mathbf{x}(v_\ell) = [\boldsymbol{\Psi}_j\mathbf{x}(v_\ell) : 1 \leq j \leq J], \quad \ell = 1, \ldots, n. \tag{2}$$

Since $2^J$ controls the maximum scale of the wavelet, in the experiments of Sec. 4 we select $J$ such that $2^J \sim \mathrm{diam}(G)$. Figure 1 plots the diffusion wavelets at different scales on two different graphs.

## 3   GEOMETRIC SCATTERING ON GRAPHS

A geometric wavelet scattering transform follows a similar construction as the (Euclidean) wavelet scattering transform of Mallat (2012), but leverages a graph wavelet transform. In this paper we utilize the wavelet transform defined in (2) of the previous section, but remark that in principle any graph wavelet transform could be used (see, e.g., Zou & Lerman, 2018). In Sec. 3.1 we define the graph scattering transform, in Sec. 3.2 we discuss its relation to other recently proposed graph scattering constructions (Gama et al., 2018; Zou & Lerman, 2018), and in Sec. 3.3 we describe several of its desirable properties as compared to other geometric deep learning algorithms on graphs.

### 3.1   GEOMETRIC SCATTERING DEFINITIONS

Machine learning algorithms that compare and classify graphs must be invariant to graph isomorphism, i.e., re-indexations of the vertices and corresponding edges. A common way to obtain invariant graph features is via summation operators, which act on a signal $\mathbf{x} = \mathbf{x}_G$ that can be defined on any graph $G$, e.g., $\mathbf{x}(v_\ell) = \deg(v_\ell)$ for each vertex $v_\ell$ in $G$. The geometric scattering transform, which is described in the remainder of this section, follows such an approach.

The simplest of such summation operators computes the sum of the responses of the signal $\mathbf{x}$. As described in Verma & Zhang (2018), this invariant can be complemented by higher order summary statistics of $\mathbf{x}$, the collection of which form statistical moments, and which are also referred to as "capsules" in that work. For example, the unnormalized $q^{\text{th}}$ moments of $\mathbf{x}$ yield the following "zero" order geometric scattering moments:

$$S\mathbf{x}(q) = \sum_{\ell=1}^{n} \mathbf{x}(v_\ell)^q, \quad 1 \leq q \leq Q \tag{3}$$

We can also replace (3) with normalized (i.e., standardized) moments of $\mathbf{x}$, in which case we store its mean ($q = 1$), variance ($q = 2$), skew ($q = 3$), kurtosis ($q = 4$), and so on. In the numerical experiments described in Sec. 4 we take $Q = 2, 3, 4$ depending upon the database, where $Q$ is chosen via cross validation to optimize classification performance. Higher order moments are not considered as they become increasingly unstable, and we report results for both normalized and unnormalized moments. In what follows we discuss the unnormalized moments, since their presentation is simpler and we use them in conjunction with fully connected layers (FCL) for classification purposes, but

the same principles also apply to normalized moments (e.g., used with SVM and logistic regression in our classification results). The invariants $S\mathbf{x}(q)$ do not capture the full variability of $\mathbf{x}$ and hence the graph $G$ upon which the signal $\mathbf{x}$ is defined. We thus complement these moments with summary statistics derived from the wavelet coefficients of $\mathbf{x}$, which in turn will lead naturally to the graph ConvNet structure of the geometric scattering transform.

Observe, analogously to the Euclidean setting, that in computing $S\mathbf{x}(1)$, which is the summation of $\mathbf{x}(v_\ell)$ over $V$, we have captured the zero frequency of $\mathbf{x}$ since $\sum_{\ell=1}^n \mathbf{x}(v_\ell) = \mathbf{x} \cdot \mathbf{1} = \sqrt{n}\,\widehat{\mathbf{x}}(0)$. Higher order moments of $\mathbf{x}$ can incorporate the full range of frequencies in $\mathbf{x}$, e.g. $S\mathbf{x}(2) = \sum_{\ell=1}^n \mathbf{x}(v_\ell)^2 = \sum_{k=1}^n \widehat{\mathbf{x}}(k)^2$, but they are mixed into one invariant coefficient. We can separate and recapture the high frequencies of $\mathbf{x}$ by computing its wavelet coefficients $\mathbf{\Psi}^{(J)}\mathbf{x}$, which were defined in (2). However, $\mathbf{\Psi}^{(J)}\mathbf{x}$ is not invariant to permutations of the vertex indices; in fact, it is covariant (or equivariant). Before summing the individual wavelet coefficient vectors $\mathbf{\Psi}_j\mathbf{x}$, though, we must first apply a pointwise nonlinearity. Indeed, define the $n \times 1$ vector $\mathbf{d}(v_\ell) = \deg(v_\ell)$, and note that $\mathbf{\Psi}_j\mathbf{x} \cdot \mathbf{d} = 0$ since one can show that $\mathbf{d}$ is a left eigenvector of $\mathbf{P}$ with eigenvalue 1. If $G$ is a regular graph then $\mathbf{d} = c\mathbf{1}$ from which it follows that $\mathbf{\Psi}_j\mathbf{x} \cdot \mathbf{1} = 0$. For more general graphs $\mathbf{d}(v_\ell) \geq 0$ for $v_\ell \in V$, which implies that for many graphs $\mathbf{1} \cdot \mathbf{d}$ will be the dominating coefficient in an expansion of $\mathbf{1}$ in an orthogonal basis containing $\mathbf{d}$; it follows that in these cases $|\mathbf{\Psi}_j\mathbf{x} \cdot \mathbf{1}| \ll 1$.

We thus apply the absolute value nonlinearity, to obtain nonlinear covariant coefficients $|\mathbf{\Psi}^{(J)}\mathbf{x}| = \{|\mathbf{\Psi}_j\mathbf{x}| : 1 \leq j \leq J\}$. We use absolute value because it is covariant to vertex permutations, non-expansive, and when combined with traditional wavelet transforms on Euclidean domains, yields a provably stable scattering transform for $q = 1$. Furthermore, initial theoretical results in Zou & Lerman (2018); Gama et al. (2018) indicate that similar graph based scattering transforms possess certain types of stability properties as well. As in (3), we extract invariant coefficients from $|\mathbf{\Psi}_j\mathbf{x}|$ by computing its moments, which define the first order geometric scattering moments:

$$S\mathbf{x}(j,q) = \sum_{\ell=1}^n |\mathbf{\Psi}_j\mathbf{x}(v_\ell)|^q, \quad 1 \leq j \leq J,\, 1 \leq q \leq Q \tag{4}$$

These first order scattering moments aggregate complimentary multiscale geometric descriptions of $G$ into a collection of invariant multiscale statistics. These invariants give a finer partition of the frequency responses of $\mathbf{x}$. For example, whereas $S\mathbf{x}(2)$ mixed all frequencies of $\mathbf{x}$, we see that $S\mathbf{x}(j,2)$ only mixes the frequencies of $\mathbf{x}$ captured by the graph wavelet $\mathbf{\Psi}_j$.

First order geometric scattering moments can be augmented with second order geometric scattering moments by iterating the graph wavelet and absolute value transforms, which leads naturally to the structure of a graph ConvNet. These moments are defined as:

$$S\mathbf{x}(j,j',q) = \sum_{i=1}^n |\mathbf{\Psi}_{j'}|\mathbf{\Psi}_j\mathbf{x}(v_i)||^q, \quad 1 \leq j < j' \leq J,\, 1 \leq q \leq Q \tag{5}$$

which consists of reapplying the wavelet transform operator $\mathbf{\Psi}^{(J)}$ to each $|\mathbf{\Psi}_j\mathbf{x}|$ and computing the summary statistics of the magnitudes of the resulting coefficients. The intermediate covariant coefficients $|\mathbf{\Psi}_{j'}|\mathbf{\Psi}_j\mathbf{x}||$ and resulting invariant statistics $S\mathbf{x}(j,j',q)$ couple two scales $2^j$ and $2^{j'}$ within the graph $G$, thus creating features that bind patterns of smaller subgraphs within $G$ with patterns of larger subgraphs (e.g., circles of friends of individual people with larger community structures in social network graphs). The transform can be iterated additional times, leading to third order features and beyond, and thus has the general structure of a graph ConvNet.

The collection of graph scattering moments $S\mathbf{x} = \{S\mathbf{x}(q),\ S\mathbf{x}(j,q),\ S\mathbf{x}(j,j',q)\}$ (illustrated in Fig. 2(a)) provides a rich set of multiscale invariants of the graph $G$. These can be used in supervised settings as input to graph classification or regression models, or in unsupervised settings to embed graphs into a Euclidean feature space for further exploration, as demonstrated in Sec. 4.

## 3.2 STABILITY AND CAPACITY OF GEOMETRIC SCATTERING

In order to assess the utility of scattering features for representing graphs, two properties have to be considered: stability and capacity. First, the stability property aims to essentially provide an upper bound on distances between similar graphs that only differ by types of deformations that can

be treated as noise. This property has been the focus of both Zou & Lerman (2018) and Gama et al. (2018), and in particular the latter shows that a diffusion scattering transform yields features that are stable to graph structure deformations whose size can be computed via the diffusion framework (Coifman & Maggioni, 2006) that forms the basis for their construction. While there are some technical differences between the geometric scattering here and the diffusion scattering in Gama et al. (2018), these constructions are sufficiently similar that we can expect both of them to have analogous stability properties. Therefore, we mainly focus here on the complementary property of the scattering transform capacity to provide a rich feature space for representing graph data without eliminating informative variance in them.

We note that even in the classical Euclidean case, while the stability of scattering transforms to deformations can be established analytically (Mallat, 2012), their capacity is typically examined by empirical evidence when applied to machine learning tasks (e.g., Bruna & Mallat, 2011; Sifre & Mallat, 2012; Andén & Mallat, 2014). Similarly, in the graph processing settings, we examine the capacity of our proposed geometric scattering features via their discriminaive power in graph data analysis tasks. In Sec. 4.1, we describe extensive numerical experiments for graph classification problems in which our scattering coefficients are utilized in conjunction with several classifiers, namely, fully connected layers (FCL, illustrated in Fig. 2(b)), support vector machine (SVM), and logistic regression. We note that SVM classification over scattering features leads to state of the art results on social network data, as well as outperforming all feed-forward neural network methods in general. Furthermore, for biochemistry data (where graphs represent molecule structures), FCL classification over scattering features outperforms all other feed-forward neural networks, even though we only train the fully connected layers. Finally, to assess the scattering feature space for data representation and exploration, in Sec. 4.2 we examine its qualities when analyzing biochemistry data, with emphasis on enzyme graphs. We show that geometric scattering enables graph embedding in a relatively low dimensional Euclidean space, while preserving insightful properties in the data. Beyond establishing the capacity of our specific construction, these results also indicate the viability of graph scattering transforms in general, as universal feature extractors on graph data, and complement the stability results established in Zou & Lerman (2018) and Gama et al. (2018).

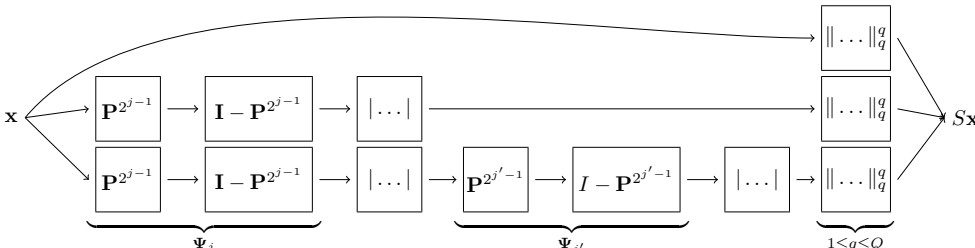

(a) Representative zeroth-, first-, and second-order cascades of the geometric scattering transform for an input graph signal $\mathbf{x}$. The presented cascades, indexed by $j, j', q$, are collected together to form the set of scattering coefficients $S\mathbf{x}$ defined in eqs. (3-5).

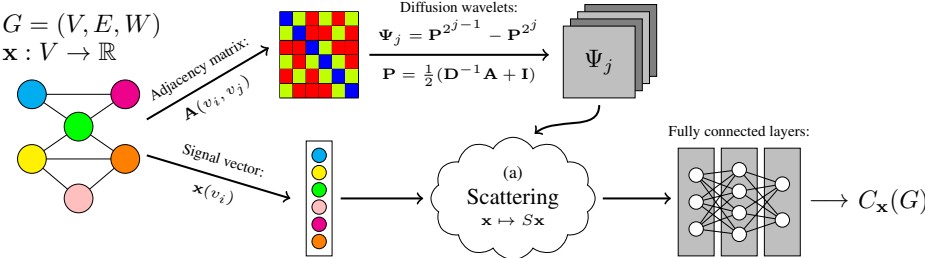

(b) Graph scattering classifier (GSC) architecture yielding class $C_{\mathbf{x}}(G)$ from graph $G$ and signal $\mathbf{x}$. The fully connected layers (FCL) can also be replaced by other classifiers (e.g., SVM or logistic regression) over scattering features, as demonstrated in Section 4.

Figure 2: Illustration of (a) the proposed scattering feature extraction (see eqs. 3, 4, and 5), and (b) its application for graph classification.

### 3.3 GEOMETRIC SCATTERING COMPARED TO OTHER FEED FORWARD GRAPH CONVNETS

We give a brief comparison of geometric scattering with other graph ConvNets, with particular interest in isolating the key principles for building accurate graph ConvNet classifiers. We begin by remarking that like several other successful graph neural networks, the graph scattering transform is covariant or equivariant to vertex permutations (i.e., commutes with them) until the final features are extracted. This idea has been discussed in depth in various articles, including Kondor et al. (2018b), so we limit the discussion to observing that the geometric scattering transform thus propagates nearly all of the information in $\mathbf{x}$ through the multiple wavelet and absolute value layers, since only the absolute value operation removes information on $\mathbf{x}$. As in Verma & Zhang (2018), we aggregate covariant responses via multiple summary statistics (i.e., moments), which are referred to there as a capsule. In the scattering context, at least, this idea is in fact not new and has been previously used in the Euclidean setting for the regression of quantum mechanical energies in Eickenberg et al. (2018; 2017) and texture synthesis in Bruna & Mallat (2018). We also point out that, unlike many deep learning classifiers (graph included), a graph scattering transform extracts invariant statistics at each layer/order. These intermediate layer statistics, while necessarily losing some information in $\mathbf{x}$ (and hence $G$), provide important coarse geometric invariants that eliminate needless complexity in subsequent classification or regression. Furthermore, such layer by layer statistics have proven useful in characterizing signals of other types (e.g., texture synthesis in Gatys et al., 2015).

A graph wavelet transform $\mathbf{\Psi}^{(J)}\mathbf{x}$ decomposes the geometry of $G$ through the lens of $\mathbf{x}$, along different scales. Graph ConvNet algorithms also obtain multiscale representations of $G$, but several works, including Atwood & Towsley (2016) and Zhang et al. (2018), propagate information via a random walk. While random walk operators like $\mathbf{P}^t$ act at different scales on the graph $G$, per the analysis in Sec. 2 we see that $\mathbf{P}^t$ for any $t$ will be dominated by the low frequency responses of $\mathbf{x}$. While subsequent nonlinearities may be able to recover this high frequency information, the resulting transform will most likely be unstable due to the suppression and then attempted recovery of the high frequency content of $\mathbf{x}$. Alternatively, features derived from $\mathbf{P}^t\mathbf{x}$ may lose the high frequency responses of $\mathbf{x}$, which are useful in distinguishing similar graphs. The graph wavelet coefficients $\mathbf{\Psi}^{(J)}\mathbf{x}$, on the other hand, respond most strongly within bands of nearly non-overlapping frequencies, each with a center frequency $k_j$ that depends on $\mathbf{\Psi}_j$.

Finally, graph labels are often complex functions of both local and global subgraph structure within $G$. While graph ConvNets are adept at learning local structure within $G$, as detailed in Verma & Zhang (2018) they require many layers to obtain features that aggregate macroscopic patterns in the graph. This is due in large part to the use of fixed size filters, which often only incorporate information from the neighbors of any individual vertex. The training of such networks is difficult due to the limited size of many graph classification databases (see Table 4 in Appendix D). Geometric scattering transforms have two advantages in this regard: (a) the wavelet filters are designed; and (b) they are multiscale, thus incorporating macroscopic graph patterns in every layer/order.

## 4 APPLICATION & RESULTS

### 4.1 GRAPH CLASSIFICATION

To evaluate the proposed geometric scattering features, we test their effectiveness for graph classification on thirteen datasets commonly used for this task. Out of these, seven datasets contain biochemistry graphs that describe molecular structures of chemical compounds, as described in the following works that introduced them: NCI1 and NCI109, Wale et al. (2008); MUTAG, Debnath et al. (1991); PTC, Toivonen et al. (2003); PROTEINS and ENZYMES, Borgwardt et al. (2005b); and D&D, Dobson & Doig (2003). In these cases, each graph has several associated vertex features $\mathbf{x}$ that represent chemical properties of atoms in the molecule, and the classification is aimed to characterize compound properties (e.g., protein types). The other six datasets, which are introduced in Yanardag & Vishwanathan (2015), contain social network data extracted from scientific collaborations (COLLAB), movie collaborations (IMDB-B & IMDB-M), and Reddit discussion threads (REDDIT-B, REDDIT-5K, REDDIT-12K). In these cases there are no inherent graph signals in the data, and therefore we compute general node characteristics (e.g., degree, eccentricity, and clustering coefficient) over them, as is considered standard practice in relevant literature (see, for example,

Verma & Zhang, 2018). A detailed description of each of these datasets appear in their respective references, and are briefly summarized in Appendix D for completeness.

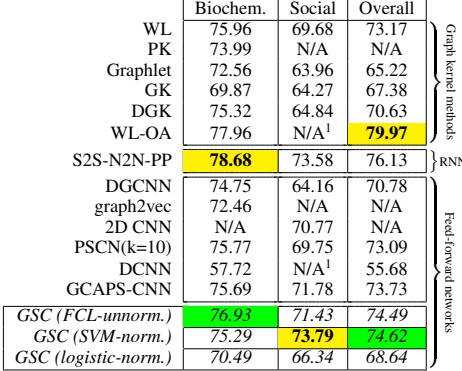
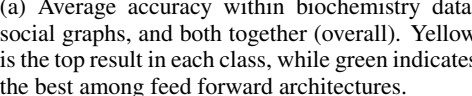

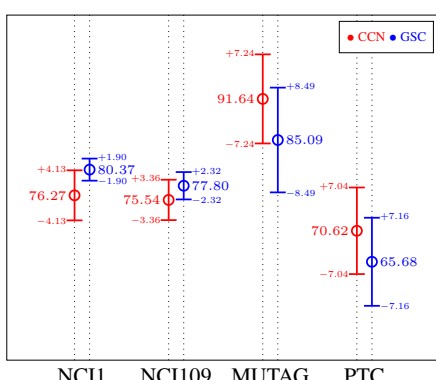

(a) Average accuracy within biochemistry data, social graphs, and both together (overall). Yellow is the top result in each class, while green indicates the best among feed forward architectures.

(b) Accuracy $\pm$ standard deviation for CCN and GSC (FCL-unnorm.) on four biochemistry datasets with reported CCN accuracy. We note than no other datasets were reported for CCN.

Figure 3: Classification accuracy (by percent correct) of the proposed method (GSC) and 14 other methods. The aggregated results in (a) are based on five biochemistry datasets and four social graph datasets. CCN is omitted from this table, as its accuracy is only reported for a handful of datasets; instead, a detailed comparison of GSC (FCL-unnorm.) with CCN is shown in (b).

In all cases, we iterate over all graphs in the database and for each one we associate graph-wide features by (1) computing the scattering features of each of the available graph signals (provided or computed), and (2) concatenating the features of all such signals. Then, the full scattering feature vectors of these graphs are passed to a classifier, which is trained from input labels, in order to infer the class for each graph. We consider three classifiers here: neural network with two/three fully connected hidden layers (FCL), SVM with RBF kernel, or logistic regression. We note that the scattering features (computed as described in Sec. 3) are based on either normalized or unnormalized moments over the entire graph. Here we used unnormalized moments for FCL, and normalized ones for other classifiers, but the difference is subtle and similar results can be achieved for the other combinations. Finally, we also note that all technical design choices for configuring our geometric scattering or the classifiers were done as part of the cross validation described in Appendix E.

We evaluate the classification results of our three geometric scattering classification (GSC) settings using ten-fold cross validation (as explained in Appendix E) and compare them to 14 prominent methods for graph classification. Out of these, six are graph kernel methods, namely: Weisfeiler-Lehman graph kernels (WL, Shervashidze et al., 2011), propagation kernel (PK, Neumann et al., 2012), Graphlet kernels (Shervashidze et al., 2009), Random walks (RW, Gärtner et al., 2003), deep graph kernels (DGK, Yanardag & Vishwanathan, 2015), and Weisfeiler-Lehman optimal assignment kernels (WL-OA, Kriege et al., 2016). Seven other methods are recent geometric feed forward deep learning algorithms, namely: deep graph convolutional neural network (DGCNN, Zhang et al., 2018), Graph2vec (Narayanan et al., 2017), 2D convolutional neural networks (2DCNN, Tixier et al., 2017), covariant compositional networks (CCN, Kondor et al., 2018a), Patchy-san (PSCN, Niepert et al., 2016, with $k = 10$), diffusion convolutional neural networks (DCNN, Atwood & Towsley, 2016), and graph capsule convolutional neural networks (GCAPS-CNN, Verma & Zhang, 2018). Finally, one method is the recently introduced recurrent neural network autoencoder for graphs (S2S-N2N-PP, Taheri et al., 2018). Following the standard format of reported classification performances for these methods (per their respective references, see also Appendix A), our results are reported in the form of average accuracy $\pm$ standard deviation (in percentages) over the ten cross-validation folds. We remark here that many of them are not reported for all datasets, and hence, we

---

[1] Accuracy for these methods was reported for less than $3/4$ of considered social graph datasets, but with biochemistry data they reach $7/9$ of all considered datasets.

mark N/A when appropriate. For brevity, the comparison is reported here in Fig. 3 in summarized form, as explained below, and in full in Appendix A.

Since the scattering transform is independent of training labels, it provides universal graph features that might not be specifically optimal in each individual dataset, but overall provide stable classification results. Further, careful examination of the results of previous methods (feed forward algorithms in particular) shows that while some may excel in specific cases, none of them achieves the best results in all reported datasets. Therefore, to compare the overall classification quality of our GSC methods with related methods, we consider average accuracy aggregated over all datasets, and within each field (i.e., biochemistry and social networks) in the following way. First, out of the thirteen datasets, classification results on four datasets (NCI109, ENZYMES, IMDB-M, REDDIT-12K) are reported significantly less frequently than the others, and therefore we discard them and use the remaining nine for the aggregation. Next, to address reported values versus N/A ones, we set an inclusion criterion of $75\%$ reported datasets for each method. This translates into at most one N/A in each individual field, and at most two N/A overall. For each method that qualifies for this inclusion criterion, we compute its average accuracy over reported values (ignoring N/A ones) within each field and over all datasets; this results in up to three reported values for each method.

The aggregated results of our GSC and 13 of the compared methods appears in Fig. 3(a). These results show that GSC (with SVM) outperforms all other methods on social network data, and in fact as shown Appendinx B, it achieves state of the art results on two datasets of this type. Additionally, the aggregated results shows that our GSC approach (with FCL or SVM) outperforms all other feed forward methods both on biochemsitry data and overall in terms of universal average accuracy[2]. The CCN method is omitted from these aggregated results, as its results in Kondor et al. (2018a) are only reported on four biochemistry datasets. For completeness, detailed comparison of GSC with this method, which appears in Fig. 3(b), shows that our method outperforms it on two datasets while CCN outperforms GSC on the other two.

## 4.2 SCATTERING FEATURE SPACE FOR DATA EXPLORATION

Geometric scattering essentially provides a task independent representation of graphs in a Euclidean feature space. Therefore, it is not limited to supervised learning applications, and can be also utilized for exploratory graph-data analysis, as we demonstrate in this section. We focus our discussion on biochemistry data, and in particular on the ENZYMES dataset. Here, geometric scattering features can be considered as providing "signature" vectors for individual enzymes, which can be used to explore interactions between the six top level enzyme classes, labelled by their Enzyme Commission (EC) numbers (Borgwardt et al., 2005a). In order to emphasize the properties of scattering-based feature extraction, rather than downstream processing, we mostly limit our analysis of the scattering feature space to linear operations such as principal component analysis (PCA).

We start by considering the viability of scattering-based embedding for dimensionality reduction of graph data. To this end, we applied PCA to our scattering coefficients (computed from unnormalized moments), while choosing the number of principal components to capture 90% explained variance. In the ENZYMES case, this yields a 16 dimensional subspace of the full scattering features space. While the Euclidean notion of dimensionality is not naturally available in the original dataset, we note that graphs in it have, on average, 124.2 edges, 29.8 vertices, and 3 features per vertex, and therefore the effective embedding of the data into $\mathbb{R}^{16}$ indeed provides a significant dimensionality reduction. Next, to verify the resulting PCA subspace still captures sufficient discriminative information with respect to classes in the data, we compare SVM classification on the resulting low dimensional vectors to the the full feature space; indeed, projection on the PCA subspace results in only a small drop in accuracy from $56.85 \pm 4.97$ (full) to $49.83 \pm 5.40$ (PCA). Finally, we also consider the dimensionality of each individual class (with PCA and $> 90\%$ exp. variance) in the scattering feature space, as we expect scattering to reduce the variability in each class w.r.t. the full feature space. In the ENZYMES case, individual classes have PCA dimensionality ranging between 6 and 10, which is indeed significantly lower than the 16 dimensions of the entire PCA space. Appendix C summarizes these findings, and repeats the described procedure for two addi-

---

[2]It should be noted, though, that if NCI109 and ENZYMES were included, the GCAPS-CNN would outperform the GSC. However, many other methods would not be comparable then.

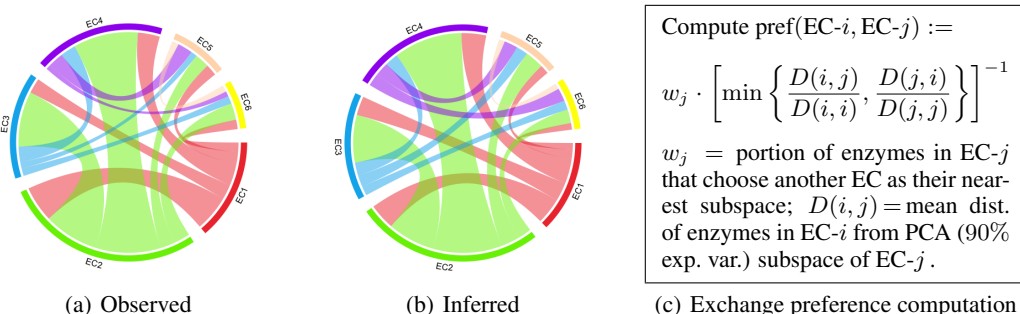

(a) Observed       (b) Inferred       (c) Exchange preference computation

Figure 4: Comparison of EC exchange preferences in enzyme evolution: (a) observed in Cuesta et al. (2015), and (b) inferred from scattering features via (c). Our inference (b) mainly recovers (a).

tional biochemistry datasets (from Wale et al., 2008) to verify that these are not unique to the specific ENZYMES dataset, but rather indicate a more general trend for geometric scattering feature spaces.

To further explore the scattering feature space, we now use it to infer relations between EC classes. First, for each enzyme $e$, with scattering feature vector $\mathbf{v}_e$ (i.e., with $S\mathbf{x}$ for all vertex features $\mathbf{x}$), we compute its distance from class EC-$j$, with PCA subspace $\mathcal{C}_j$, as the projection distance: $\mathrm{dist}(e, \mathrm{EC}\text{-}j) = \|\mathbf{v}_e - \mathrm{proj}_{\mathcal{S}_j} \mathbf{v}_e\|$. Then, for each enzyme class EC-$i$, we compute the mean distance of enzymes in it from the subspace of each EC-$j$ class as $D(i,j) = \mathrm{mean}\{\mathrm{dist}(e, \mathrm{EC}\text{-}j) : e \in \mathrm{EC}\text{-}i\}$. Appendix C summarizes these distances, as well as the proportion of points from each class that have their true EC as their nearest (or second nearest) subspace in the scattering feature space. In general, $48\%$ of enzymes select their true EC as the nearest subspace (with additional $19\%$ as second nearest), but these proportions vary between individual EC classes. Finally, we use these scattering-based distances to infer EC exchange preferences during enzyme evolution, which are presented in Fig. 4 and validated with respect to established preferences observed and reported in Cuesta et al. (2015). We note that the result there is observed independently from the ENZYMES dataset. In particular, the portion of enzymes considered from each EC is different between these data, since Borgwardt et al. (2005b) took special care to ensure each EC class in ENZYMES has exactly 100 enzymes in it. However, we notice that in fact the portion of enzymes (in each EC) that choose the wrong EC as their nearest subspace, which can be considered as EC "incoherence" in the scattering feature space, correlates well with the proportion of evolutionary exchanges generally observed for each EC in Cuesta et al. (2015), and therefore we use these as EC weights in Fig. 4(c). Our results in Fig. 4 demonstrate that scattering features are sufficiently rich to capture relations between enzyme classes, and indicate that geometric scattering has the capacity to uncover descriptive and exploratory insights in graph data analysis, beyond the supervised graph classification from Sec 4.1.

## 5    CONCLUSION

We presented the geometric scattering transform as a deep filter bank for feature extraction on graphs. This transform generalizes the scattering transform, and augments the theoretical foundations of geometric deep learning. Further, our evaluation results on graph classification and data exploration show the potential of the produced scattering features to serve as universal representations of graphs. Indeed, classification with these features with relatively simple classifier models reaches high accuracy results on most commonly used graph classification datasets, and outperforms both traditional and recent deep learning feed forward methods in terms of average classification accuracy over multiple datasets. We note that this might be partially due to the scarcity of labeled big data in this field, compared to more traditional ones (e.g., image or audio classification). However, this trend also correlates with empirical results for the classic scattering transform, which excels in cases with low data availability. Finally, the geometric scattering features provide a new way for computing and considering global graph representations, independent of specific learning tasks. Therefore, they raise the possibility of embedding entire graphs in Euclidean space and computing meaningful distances between graphs with them, which can be used for both supervised and unsupervised learning, as well as exploratory analysis of graph-structured data.

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

## APPENDIX A  FULL COMPARISON TABLE

Table 1: Comparison of the proposed graph scattering classifier (GSC) with graph kernel methods and deep learning methods on biochemistry & social graph datasets.

| Type | Method | NCI1 | NCI109 | D&D | PROTEINS | MUTAG | PTC | ENZYMES |
|---|---|---|---|---|---|---|---|---|
| Graph kernel methods | WL | 84.46 ± 0.45 | 85.12 ± 0.29 | 78.34 ± 0.62 | 72.92 ± 0.56 | 84.11 ± 1.91 | 59.97 ± 1.60 | 55.22 ± 1.26 |
| Graph kernel methods | PK | 82.54 ± 0.47 | N/A | 78.25 ± 0.51 | 73.68 ± 0.68 | 76.00 ± 2.69 | 59.50 ± 2.44 | N/A |
| Graph kernel methods | Graphlet | 70.5 ± 0.2 | 69.3 ± 0.2 | 79.7 ± 0.7 | 72.7 ± 0.6 | 85.2 ± 0.9 | 54.7 ± 2.0 | 30.6 ± 1.2 |
| Graph kernel methods | WL-OA | 86.1 ± 0.2 | 86.3 ± 0.2 | 79.2 ± 0.4 | 76.4 ± 0.4 | 84.5 ± 1.7 | 63.6 ± 1.5 | 59.9 ± 1.1 |
| Graph kernel methods | GK | 62.28 ± 0.29 | 62.60 ± 0.19 | 78.45 ± 0.26 | 71.67 ± 0.55 | 81.39 ± 1.74 | 57.26 ± 1.41 | 26.61 ± 0.99 |
| Graph kernel methods | DGK | 80.3 ± 0.4 | 80.3 ± 0.3 | 73.09 ± 0.25 | 75.7 ± 0.50 | 87.4 ± 2.7 | 60.1 ± 2.5 | 53.4 ± 0.9 |
| RNN | S2S-P2P-NN | 83.72 ± 0.4 | 83.64 ± 0.3 | N/A | 76.61 ± 0.5 | 89.86 ± 1.1 | 64.54 ± 1.1 | 63.96 ± 0.6 |
| Feed-forward networks | DGCNN | 74.44 ± 0.47 | N/A | 79.37 ± 0.94 | 75.54 ± 0.94 | 85.83 ± 1.66 | 58.59 ± 2.47 | 51.00 ± 7.29 |
| Feed-forward networks | graph2vec | 73.22 ± 1.81 | 74.26 ± 1.47 | N/A | 73.30 ± 2.05 | 83.15 ± 9.25 | 60.17 ± 6.86 | N/A |
| Feed-forward networks | 2D CNN | N/A | N/A | N/A | 77.12 ± 2.79 | N/A | N/A | N/A |
| Feed-forward networks | CCN | 76.27 ± 4.13 | 75.54 ± 3.36 | N/A | N/A | 91.64 ± 7.24 | 70.62 ± 7.04 | N/A |
| Feed-forward networks | PSCN ($k=10$) | 76.34 ± 1.68 | N/A | 76.27 ± 2.15 | 75.00 ± 2.51 | 88.95 ± 4.37 | 62.29 ± 5.68 | 42.44 ± 1.76 |
| Feed-forward networks | DCNN | 56.61 ± 1.04 | 57.47 ± 1.22 | 58.09 ± 0.53 | 61.29 ± 1.60 | 56.60 ± 2.89 | 56[3] | 61.83 ± 5.39 |
| Feed-forward networks | GCAPS-CNN | 82.72 ± 2.38 | 81.12 ± 1.28 | 77.62 ± 4.99 | 76.40 ± 4.17 | N/A | 66.01 ± 5.91 | N/A |
| | GSC (unnorm.) | 80.37 ± 1.90 | 77.80 ± 2.32 | 78.86 ± 3.72 | 74.67 ± 2.90 | 85.09 ± 8.49 | 65.68 ± 7.16 | 53.33 ± 4.94 |
| | GSC (SVM-norm.) | 78.56 ± 2.49 | 76.98 ± 1.88 | 75.04 ± 3.64 | 75.03 ± 5.05 | 83.57 ± 6.75 | 64.24 ± 3.96 | 56.83 ± 4.97 |
| | GSC (logistic-norm.) | 69.76 ± 2.65 | 68.50 ± 2.45 | 75.30 ± 2.60 | 72.42 ± 3.23 | 72.72 ± 11.73 | 62.23 ± 6.65 | 38.67 ± 7.77 |

| Type | Method | COLLAB | IMDB-B | IMDB-M | REDDIT-B | REDDIT-5K | REDDIT-12K |
|---|---|---|---|---|---|---|---|
| Graph kernel | WL | 77.82 ± 1.45 | 71.60 ± 5.16 | N/A | 78.52 ± 2.01 | 50.77 ± 2.02 | 34.57 ± 1.32 |
| Graph kernel | PK | N/A | N/A | N/A | N/A | N/A | N/A |
| Graph kernel | Graphlet | 73.42 ± 2.43 | 65.4 ± 5.95 | N/A | 77.26 ± 2.34 | 39.75 ± 1.36 | 25.98 ± 1.29 |
| Graph kernel | WL-OA | 80.7 ± 0.1 | N/A | N/A | 89.3 ± 0.3 | N/A | N/A |
| Graph kernel | GK | 72.84 ± 0.28 | 65.87 ± 0.98 | 43.89 ± 0.38 | 77.34 ± 0.18 | 41.01 ± 0.17 | N/A |
| Graph kernel | DGK | 73.0 ± 0.2 | 66.9 ± 0.5 | 44.5 ± 0.5 | 78.0 ± 0.3 | 41.2 ± 0.1 | 32.2 ± 0.1 |
| RNN | S2S-P2P-NN | 81.75 ± 0.8 | 73.8 ± 0.7 | 51.19 ± 0.5 | 86.50 ± 0.8 | 52.28 ± 0.5 | 42.47 ± 0.1 |
| Feed-forward networks | DGCNN | 73.76 ± 0.49 | 70.03 ± 0.86 | 47.83 ± 0.85 | N/A | 48.70 ± 4.54 | N/A |
| Feed-forward networks | graph2vec | N/A | N/A | N/A | N/A | N/A | N/A |
| Feed-forward networks | 2D CNN | 71.33 ± 1.96 | 70.40 ± 3.85 | N/A | 89.12 ± 1.7 | 52.21 ± 2.44 | 48.13 ± 1.47 |
| Feed-forward networks | CCN | N/A | N/A | N/A | N/A | N/A | N/A |
| Feed-forward networks | PSCN ($k=10$) | 72.60 ± 2.15 | 71.00 ± 2.29 | 45.23 ± 2.84 | 86.30 ± 1.58 | 49.10 ± 0.7 | 41.32 ± 0.42 |
| Feed-forward networks | DCNN | 52.11 ± 0.71 | 49.06 ± 1.37 | 33.49 ± 1.42 | N/A | N/A | N/A |
| Feed-forward networks | GCAPS-CNN | 77.71 ± 2.51 | 71.69 ± 3.40 | 48.50 ± 4.1 | 87.61 ± 2.51 | 50.10 ± 1.72 | N/A |
| | GSC (unnorm.) | 76.50 ± 1.20 | 71.30 ± 2.87 | 47.73 ± 4.42 | 86.30 ± 2.56 | 51.63 ± 2.08 | 38.39 ± 1.19 |
| | GSC (SVM-norm.) | 80.02 ± 1.63 | 71.90 ± 3.45 | 48.27 ± 4.19 | 89.65 ± 1.94 | 53.57 ± 2.42 | 45.23 ± 1.25 |
| | GSC (logistic-norm.) | 72.64 ± 2.27 | 63.70 ± 3.69 | 41.53 ± 3.50 | 80.60 ± 2.22 | 48.41 ± 3.41 | N/A |

All results come from the respective papers that introduced the methods, with the exception of: (1) social network results of WL, from Tixier et al. (2017); (2) biochemistry and social results of DCNN, from Verma & Zhang (2018); (3) biochemistry, except for D&D, and social result of GK,

---

[3]DCNN using different training/test split

from Yanardag & Vishwanathan (2015); (4) D&D of GK is from Niepert et al. (2016); and (5) for Graphlets, biochemistry results from Kriege et al. (2016), social results from Tixier et al. (2017).

## APPENDIX B    STATE OF THE ART RESULTS ON REDDIT DATASETS

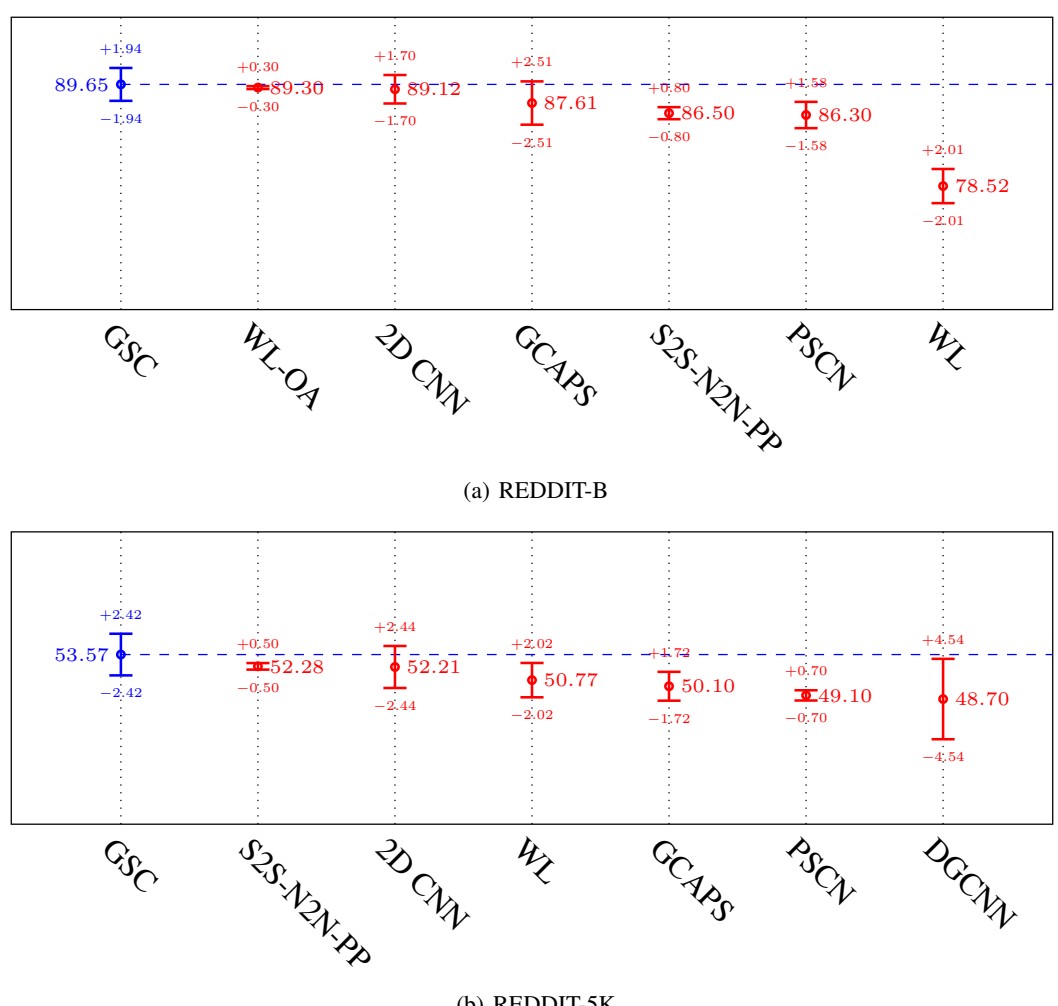

(a) REDDIT-B

(b) REDDIT-5K

Figure 5: Accuracy ± standard deviation for GSC (SVM-norm.) and top six other methods on two REDDIT datasets. GSC achieves state of the art accuracy on these datasets.

## APPENDIX C    DETAILED TABLES FOR SCATTERING FEATURE SPACE ANALYSIS FROM SECTION 4.2

Table 2: Dimensionality reduction with PCA over scattering features (unnorm. moments)

| Dataset | SVM accuracy | | PCA dimensions (> 90% variance) | | | | | | | |
|---|---|---|---|---|---|---|---|---|---|---|
| | PCA | Full | All classes | Per class | | | | | | |
| ENZYMES | $49.83 \pm 5.40$ | $56.83 \pm 4.97$ | 16 | 9 | 8 | 8 | 9 | 10 | 6 |
| NCI1 | $73.84 \pm 2.58$ | $79.12 \pm 2.21$ | 39 | 29 | | | 26 | | |
| NCI109 | $72.04 \pm 2.28$ | $77.83 \pm 1.61$ | 39 | 32 | | | 27 | | |

Table 3: EC subspace analysis in scattering feature space of ENZYMES (Borgwardt et al., 2005a)

| Enzyme Class: | Mean distance to subspace of class | | | | | | True class as | | |
|---|---|---|---|---|---|---|---|---|---|
| | EC-1 | EC-2 | EC-3 | EC-4 | EC-5 | EC-6 | 1st | 2nd | 3rd-6th |
| | measured via PCA projection/reconstruction distance | | | | | | nearest subspace | | |
| EC-1 | 18.15 | 98.44 | 75.47 | 62.87 | 53.07 | 84.86 | 45% | 28% | 27% |
| EC-2 | 22.65 | 9.43 | 30.14 | 22.66 | 18.45 | 22.75 | 53% | 24% | 23% |
| EC-3 | 107.23 | 252.31 | 30.4 | 144.08 | 117.24 | 168.56 | 32% | 7% | 61% |
| EC-4 | 117.68 | 127.27 | 122.3 | 29.59 | 94.3 | 49.14 | 24% | 12% | 64% |
| EC-5 | 45.46 | 66.57 | 60 | 50.07 | 15.09 | 58.22 | 67% | 21% | 12% |
| EC-6 | 62.38 | 58.88 | 73.96 | 51.94 | 59.23 | 13.56 | 67% | 21% | 12% |

## APPENDIX D    DETAILED DATASET DESCRIPTIONS

The details of the datasets used in this work are as follows (see the main text in Sec. 3 for references):

**NCI1** contains 4,110 chemical compounds as graphs, with 37 node features. Each compound is labeled according to is activity against non-small cell lung cancer and ovarian cancer cell lines, and these labels serve as classification goal on this data.

**NCI109** is similar to NCI1, but with 4,127 chemical compounds and 38 node features.

**MUTAG** consists of 188 mutagenic aromatic and heteroaromatic nitro compounds (as graphs) with 7 node features. The classification here is binary (i.e., two classes), based on whether or not a compound has a mutagenic effect on bacterium.

**PTC** is a dataset of 344 chemical compounds (as graphs) with nineteen node features that are divided into two classes depending on whether they are carcinogenic in rats.

**PROTEINS** dataset contains 1,113 proteins (as graphs) with three node features, where the goal of the classification is to predict whether the protein is enzyme or not.

**D&D** dataset contains 1,178 protein structures (as graphs) that, similar to the previous one, are classified as enzymes or non-enzymes.

**ENZYMES** is a dataset of 600 protein structures (as graphs) with three node features. These proteins are divided into six classes of enzymes (labelled by enzyme commission numbers) for classification.

**COLLAB** is a scientific collaboration dataset contains 5K graphs. The classification goal here is to predict whether the graph belongs to a subfield of Physics.

**IMDB-B** is a movie collaboration dataset with contains 1K graphs. The graphs are generated on two genres: Action and Romance, the classification goal is to predict the correct genre for each graph.

**IMDB-M** is similar to IMDB-B, but with 1.5K graphs & 3 genres: Comedy, Romance, and Sci-Fi.

**REDDIT-B** is a dataset with 2K graphs, where each graph corresponds to an online discussion thread. The classification goal is to predict whether the graph belongs to a Q&A-based community or discussion-based community.

**REDDIT-5K** consists of 5K threads (as graphs) from five different subreddits. The classification goal is to predict the corresponding subreddit for each thread.

**REDDIT-12K** is similar to REDDIT-5k, but with 11,929 graphs from 12 different subreddits.

Table 4 summarizes the size of available graph data (i.e., number of graphs, and both max & mean number of vertices within graphs) in these datasets, as previously reported in the literature.

**Graph signals for social network data:**   None of the social network datasets has ready-to-use node features. Therefore, in the case of COLLAB, IMDB-B, and IMDB-M, we use the eccentricity, degree, and clustering coefficients for each vertex as characteristic graph signals. In the case of REDDIT-B, REDDIT-5K and REDDIT-12K, on the other hand, we only use degree and clustering coefficient, due to presence of disconnected graphs in these datasets.

|  | NCI1 | NCI109 | MUTAG | D&D | PTC | PROTEINS |
|---|---|---|---|---|---|---|
| # of graphs in data: | 4110 | 4127 | 188 | 1178 | 344 | 1113 |
| Max # of vertices: | 111 | 111 | 28 | 5748 | 109 | 620 |
| Mean # of vertices: | 29.8 | 29.6 | 17.93 | 284.32 | 25.56 | 39.0 |
| # of features per vertex: | 37 | 38 | 7 | 89 | 22 | 3 |
| Mean # of edges: | 64.6 | 62.2 | 39.50 | 1431.3 | 51.90 | 72.82 |
| # of classes: | 2 | 2 | 2 | 2 | 2 | 2 |

| ENZYMES | COLLAB | IMDB | | REDDIT | | |
|---|---|---|---|---|---|---|
|  |  | B | M | B | 5K | 12K |
| 600 | 5000 | 1000 | 1500 | 2000 | 5000 | 11929 |
| 126 | 492 | 136 | 89 | 3783 | 3783 | 3782 |
| 32.6 | 74.49 | 19.77 | 13 | 429.61 | 508.5 | 391.4 |
| 3 | 3 | 3 | 3 | 2 | 2 | 2 |
| 124.2 | 2457.78 | 96.53 | 65.94 | 497.75 | 594.87 | 456.89 |
| 6 | 3 | 2 | 3 | 2 | 5 | 11 |

Table 4: Basic statistics of the graph classification databases

## APPENDIX E  TECHNICAL DETAILS

The computation of the scattering features described in Section 3 is based on several design choices, akin to typical architecture choices in neural networks. Most importantly, it requires a choice of 1. which statistical moments to use (normalized or unnormalized), 2. the number of wavelet scales to use (given by $J$), and 3. the number of moments to use (denoted by $Q$). The configuration used for each dataset in this work is summarized in Table 5, together with specific settings used in the downstream classification layers, as descibed below.

Once the scattering coefficients are generated through the above processes, they are either fed into a standard classifier (SVM or logistic regression), or into two or three fully connected layers (see Table 5 for specifics) and then a softmax layer that is used to compute the class probabilities. In the latter case, cross entropy loss is minimized during the training process and ReLU is used as the activation function between fully connected layers. Besides, we use mini batch training with batch size 64 and ADAM optimization technique for training. Two learning rates 0.002 and 0.02 are tested during training. Optimal training epochs are decided through cross validation. Finally, $L_2$ norm regularization is used to avoid overfittings.

**Cross validation procedure:**  Classification evaluation was done with standard ten-fold cross validation procedure. First, the entire dataset is randomly split into ten subsets. Then, in each iteration (or "fold"), nine of them are used as training and validation, and the other one is used for testing classification accuracy. In total, after ten iterations, each of the subsets has been used once for testing, resulting in ten reported classification accuracy numbers for the examined dataset. Finally, the mean and standard deviation of these ten accuracies are computed and reported.

It should be noted that when using fully connected layers, each iteration also performs automatic tuning of the trained classifier, as follows. First, nine iterations are performed, each time using eight subsets (i.e., folds) as training and the remaining one as validation set, which is used to determine the optimal epoch for network training. Then, the classifier is retrained with all nine subsets. After nine iterations, each of the training/validation subsets has been used once for validation, and we obtain nine classification models, which in turn produce nine predictions (i.e., class probabilities) for each data point in the test subset of the main cross validation. To obtain the final result of this cross validation iteration, we sum up all these predictions and select the class with the highest probability as our final classification result. These results are then compared to the true labels (in the test set) on the test subset to obtain classification accuracy for this fold.

| Database | Scattering | | | Fully connected | | |
| | Moment | $J$ | $Q$ | # Hidden units 1 | # Hidden units 2 | # Hidden units 3 |
|---|---|---|---|---|---|---|
| NCI1 | un-normalized | 5 | 3 | 40 | 20 | 0 |
| | normalized | 4 | 3 | 60 | 30 | 0 |
| NCI109 | un-normalized | 5 | 4 | 60 | 30 | 15 |
| | normalized | 5 | 3 | 60 | 30 | 15 |
| D&D | un-normalized | 5 | 2 | 20 | 0 | 0 |
| | normalized | 5 | 4 | 20 | 0 | 0 |
| PROTEINS | un-normalized | 5 | 3 | 20 | 10 | 0 |
| | normalized | 4 | 3 | 20 | 10 | 5 |
| MUTAG | un-normalized | 4 | 4 | 40 | 20 | 0 |
| | normalized | 5 | 4 | 60 | 0 | 0 |
| PTC | un-normalized | 5 | 4 | 50 | 20 | 0 |
| | normalized | 5 | 3 | 50 | 25 | 0 |
| ENZYMES | un-normalized | 4 | 4 | 60 | 30 | 15 |
| | normalized | 5 | 4 | 60 | 30 | 15 |
| COLLAB | un-normalized | 4 | 3 | 60 | 30 | 0 |
| | normalized | 5 | 4 | 50 | 20 | 0 |
| IMDB-B | un-normalized | 4 | 4 | 50 | 20 | 10 |
| | normalized | 4 | 3 | 50 | 20 | 10 |
| IMDB-M | un-normalized | 4 | 4 | 50 | 20 | 0 |
| | normalized | 5 | 3 | 60 | 30 | 0 |
| REDDIT-B | un-normalized | 5 | 3 | 60 | 30 | 15 |
| | normalized | 5 | 4 | 60 | 30 | 15 |
| REDDIT-5K | un-normalized | 5 | 3 | 50 | 20 | 10 |
| | normalized | 4 | 4 | 60 | 30 | 15 |
| REDDIT-12K | un-normalized | 5 | 4 | 100 | 50 | 25 |
| | normalized | 5 | 4 | 100 | 50 | 25 |

Table 5: Settings of the geometric scattering classifier

**Software & hardware environment:** Geometric scattering and related classification code were implemented in Python with TensorFlow. All experiments were performed on HPC environment using an intel16-k80 cluster, with a job requesting one node with four processors and two Nvidia Tesla k80 GPUs.

