# OpenReview forum: "Graph Classification with Geometric Scattering"
_ICLR.cc/2019/Conference_

### Official Review · AnonReviewer1 · 2018-10-28
**A good idea, but the delineation to other work needs improvement**

**Rating:** 5
**Confidence:** 4

**Review:**

# Summary of the paper

Inspired by the success of deep filter banks, this paper presents a designed deep filter bank for graphs that is based on random walks.  More precisely, the technique uses lazy random walks, expressed in terms of the graph Laplacian, and re-frames this in terms of graph signal processing. Similarly to wavelets, graph node features are calculated at different scales and subsequently summed in order to remain invariant under permutations. Several experiments on graph data sets demonstrate the performance of the new technique.

# Review

This paper is written very well and explains its method with high clarity. The principal issues I see are as follows:

- The originality of the contributions is not clear
- Missing theoretical discussion
- The experimental setup is terse and slightly confusing

Concerning the originality of the paper, the differences to Gama et al., 'Diffusion Scattering Transforms on Graphs' are not made clear. Cursory reading of this publication shows a large degree of similarity. Both of the papers make use of diffusion geometry, but Gama et al. _also_ define a multi-scale filter bank, similar to Eq. 4 and 5. The paper needs to position itself more clearly vis-à-vis this other publication. Is the present approach to be seen more as an application of the theory that was developed in the paper by Gama et al.? What are the key similarities and differences? In terms of space, this could be added to Section 3.2, which could be rephrased as a generic 'Differences to other methods' section and has to be slightly condensed in any case (see my suggestions below). Another publication by Zou & Lerman, 'Graph Convolutional Neural Networks via Scattering', is also cited as an inspiration, but here the differences are larger in my understanding and do not necessitate further justification. Last, the publication 'Graph Capsule Convolutional Neural Networks' by Verma & Zhang is also cited for the definition of 'scattering capsules'. Again, cursory reading of the publication shows that this approach is similar to the presented one; the only difference being which features are used for the definition of capsules. I recommend referring to the invariants as 'capsules' and link it back to Verma & Zhang so that the provenance of the terminology is clear.

Concerning the theoretical part of the paper, I miss a discussion of the complexity of the approach. Such a discussion does not have to be long, but in particular since the paper mentions that the applicability of scattering transforms for transfer learning (and also remarks about the universality of them in Section 4), some space should be devoted to theoretical considerations (memory complexity, runtime complexity). This would strengthen the paper a lot, in particular in light of the complexity of other approaches! Furthermore, an additional experiment about the stability of scattering transforms appears warranted. While I applaud the experimental description in the paper (number of scales, how the maximum scale is chosen, ...), an additional proof or experiment in the appendix should deal with the stability. Let's assume that for extremely large graphs, I am content with 'almost-but-not-quite-as-good' classification performance. Is it possible to achieve this by limiting the number of scales? How much to the results depend on the 'right' choice here?

Concerning the experimental setup, I think that the way (average) accuracies are reported at present is slightly misleading. The paper even remarks about this in footnote 2. While I understand the need of demonstrating the universality of these features, I think that the current setup is not optimal for this. I would recommend (in addition to reporting accuracies) a transfer learning setup rather in which the beneficial properties of the new method can be better explored. More precisely, the claim from Section 4, 4th paragraph ('Since the scattering transform...') needs to be further explored. This appears to be a unique feature of the new method. The current experimental setup does not exploit it. As a side-note, I realize that this might sound like a standard request for 'show more experiments', but I think the paper would be more impactful if it contained one scenario in which its benefits over other approaches are clear.

# Suggestions for improvement

The paper flows extremely well and it is clear that care has been taken to ensure that everything can be understood. I liked the discussion of invariance properties in particular. There are only a few minor things that can be improved:

- 'covariant' and 'equivariant', while common in (graph) signal processing, could be briefly explained to increase accessibility and impact
- 'order' and 'layer' are not used consistently: in the caption of Figure 2a, the term 'order' is used, but for Eq. 4 and 5, for example, the term 'layer' is employed. Since 'layer' is more reminiscent of a DNN, I would suggest to use 'order' throughout the paper, because it meshes better with the way the scattering invariants are defined.
- the notation $Sx$ is slightly overloaded; in Figure 2a, for example, it is not clear at first that the individual cascades are supposed to form a *set*; this is only explained at the end of Section 3.1; to make matters more consistent, the figure should be updated and the combination of individual cascades should be made clear
- In Eq. 5, the bars of the absolute value are not set correctly; the absolute value should cover $\psi_j x(v_i)$ and not $(v_i)$ itself.
- minor 'gripe': $\psi^{(J)}$ is defined as a set in Eq. 2, but it is treated as a matrix or an operator (and also referred to as such); this should be more consistent
- The discussion of the aggregation of multiple statistics in Section 3.2 appears to be somewhat redundant in light of the discussion for Eq. 4 and Eq. 5 in the preceding section
- in the appendix, more details about the training of the FCN should be added; all other parts of the experiments are described in sufficient detail, but the training process requires additional information about learning rates etc.

---

> ### Author Response · Authors · 2018-11-26
> **Thank you for the detailed and helpful comments! Let us try to respond point by point.**
>
> Thank you for the detailed and helpful comments! Let us try to respond point by point here and in the subsequent (part II) post.
>
> Regarding the differences and similarities with [Gama, et al, 2018], we agree that the construction presented here is similar to [Gama, et al, 2018], and as the reviewer suggested, we added a new subsection (in section 3) to discuss the relation between our work and that one, since we consider them complementary to each other. We point out that while [Gama, et al, 2018] prove some very nice theoretical properties of their diffusion scattering transform, their numerical experiments do not give much indication as to whether this theory is practically relevant to graph learning tasks (supervised or unsupervised). In this paper we have shown that, indeed, they potentially are - especially with new results that we now added, both for graph classification (using SVM over scattering features - achieving state of the art results on REDDIT B and REDDIT 5K, and in aggregate over social networks) and for data exploration, as we describe below. Furthermore, both [Zou and Lerman, 2018] and [Gama, et al, 2018] focus primarily on stability results, i.e., if two graphs are similar (e.g., of the same class), will the resulting scattering coefficients also be similar? Under certain theoretical frameworks, they prove positive statements along these lines. Some of the new experiments we have added in (see below), though, aim at shedding light on the converse to this question, namely: if two graphs are dissimilar (e.g., in different classes), are the geometric scattering coefficients sufficiently different to separate the classes? While we don’t prove any theoretical results along these lines, new numerical experiments that we added in the revised version of the paper are sufficiently positive that they open up this question for further numerical and theoretical work. Therefore, as we mention above, we view this paper as complementary to the work of [Gama, et al, 2018] (as well as [Zou and Lerman, 2018]), as it both fills in missing numerical validation and potentially opens up a new path for theoretical investigation. We also note that even in the classical case, the stability of the scattering transform is established with rigorous mathematical analysis, while the capacity of scattering features to obtain rich representations of signals is established by applications and numerical experiments. We follow a similar approach here.
>
> For the capsule graph neural networks of [Verma and Zhang, 2018], we would argue that the capsule part is not the primary theoretical addition; indeed, capsule in the context presented in our paper and in [Verma and Zhang, 2018] is just another name for statistical moments (and in fact in the updated version of the paper, we are switching to this terminology, while still linking to [Verma and Zhang, 2018]), which people have been computing and using in various contexts for a very long time. It is therefore the features themselves that constitute the most important difference, and here indeed there are several differences. Specifically, [Verma and Zhang, 2018] like other graph CNN methods, learn the graph convolutional filters in a supervised, task driven fashion. Furthermore, they use filters with a fixed scale, which in fact they acknowledge as a shortcoming, but since they are learned they refrain from training larger (multiscale) filters. Rather, they define global input features to their graph convolutional neural network. On the other hand, the geometric scattering transform uses wavelet graph filters that adapt to the graph structure, but not the classification task, and they are multiscale.
>
> Regarding the computational complexity: the transform consists of a sequence of standard matrix multiplications, absolute value operators, and summations. We point out that since the graph wavelet filters are not learned, the training time is completely determined the time needed to train the fully connected layers or the SVM, which is not much and we believe is well understood independently of the proposed method here. In particular this means that unlike other geometric deep learning methods, the graph convolution matrix multiplications do not need to be repeated numerous times in the training process, but rather are a one time computational cost that is carried out before training. Further, in the added experiments (Sec. 4.2), we also indicate that the scattering features can in fact be used for dimensionality reduction by embedding graph data into a low dimensional Euclidean space without losing much in terms of classification accuracy compared to the full scattering feature space.
>
> --- continued in the following post ---

---

> > ### Author Response · Authors · 2018-11-26
> > **Part II of our response**
> >
> > Regarding experimental setup/results: we note that in the revised version we added classification with RBF kernel SVM, which outperforms all other methods on two individual datasets (REDDIT B and REDDIT 5K), in addition to outperforming all other methods on average on social network data. This also reduces the dependence of the presentation on individual classifiers, and indicates the scattering feature space is sufficiently rich to enable state of the art classification even without the aggregation of classification accuracy across datasets, which as we explain in the paper, is intended to assess the universality of our features.
> >
> > Regarding the stability of the transform vis a vis the parameters choices, we thought this was a good idea, but amended it slightly. We instead looked at the stability of the classification results in terms of the intrinsic dimension of the scattering coefficients, i.e., if we compute the PCA projection of the scattering coefficients and only keep a number of dimensions that is necessary to capture 90% of the variance of the scattering coefficients, how does this affect the classification performance? We see that, first, according to this measure of dimensionality, the geometric scattering coefficients give intrinsically low dimensional descriptions of the data. It is likely that other graph CNNs do the same, but we again emphasize that the graph convolution filters in these networks are task dependent, whereas the geometric scattering wavelet filters are task independent and more suitable for exploratory data analysis. Furthermore, using only the PCA projections (capturing 90% of the variance of the geometric scattering coefficients), we carried out an SVM classification and found the classification rate does not drastically decrease relative to an SVM classification on the full set of geometric scattering coefficients. In particular, the geometric scattering coefficients aggregate useful information into the primary dimensions of variability, and indeed these dimensions describe the majority of class differences.
> >
> > To further establish the utility of geometric scattering as a unique feature extraction method beyond graph classification, we added new experiments that look into data exploration with it, focusing on the ENZYMES dataset. Here, we indeed show that scattering features enable inference of enzyme commission class exchanges in the evolution of enzymes, which emerge in an unsupervised manner from our scattering feature space, and are validate with established knowledge from [Cuesta et al., 2015]. We note that the revised version of the paper also contains improvements in the classification performances with the addition of SVM based classification, which also serves to demonstrate the independence of the geometric scattering transform itself from specific classifiers, as we now show it with three classifiers (SVM, fully-connected layers, or logistic regression).
> >
> > Regarding the other (more minor) comments: we did our best to address them as part of this revision.
> >
> > We hope that with these additions we were able to sufficiently improve the quality of the presented results to warrant an update to the reviewer's score.

---

> > > ### Comment · AnonReviewer1 · 2018-12-03
> > > **Thank you**
> > >
> > > Thanks for the extensive changes of the paper! I appreciate the work that you did, but I am still not convinced about the novelty of the approach as well as its practical benefits. The 'edge' over existing methods does not appear to be sufficiently large to me.
> > >
> > > With the new data sets, the difference over existing work is not so large (as far as I understand, it is within the standard error that you calculate); for the REDDIT-5k data set, a previous approach that is geared towards using topological information (https://papers.nips.cc/paper/6761-deep-learning-with-topological-signatures.pdf) outperforms your proposed method. For the REDDIT-B data set, the WL-OA kernel, as you correctly cite it, is on a par with your approach considering its smaller standard deviation.
> > >
> > > I think this paper would benefit from a large-scale investigation to _really_ drive the point home about the benefits of the methods.

---

### Official Review · AnonReviewer3 · 2018-10-31
**Interesting paper and ideas, a bit low on results maybe**

**Rating:** 6
**Confidence:** 3

**Review:**

This paper generalizes scattering transform to graphs. It defines wavelets, scattering coefficients on graph signals. The experimental section describes their use in classification tasks with comparisons with recent methods. It seems scattering performs less well than SOTA methods, but has the advantages of not requiring any training so potentially good candidates for low data regimes application. Interesting and original paper and ideas being developed, but might be a tiny bit weak in term of results, both theoretical and experimental ?

There is not much theoretical results (mostly definition and hints that some of the results from euclidian case might generalize without formal investigation).

Regarding the results, in particular table3, given that you use particular hyper parameters J and Q, for each dataset, this is arguably a bit of architectural overfitting ? Results would be more convincing IMO if obtained with a single set of hyper parameters. What was the procedure to come up with those parameters ?

Regarding the methodology for training the classifier, I am not familiar with these datasets but using just a 1/10 of the data to train classifier seems a bit extreme ?
How about training each on 90% random subset of training set and averaging ? Or just the whole training subset ? That would still be fine in the sense that none of the classifier would have seen the test set ?

p2 '~it naturally extends to multiple signals by concatenating their scattering features~'

P4 figure 1: Not very clear what those visualizations are. \Psi_j is supposedly a n x n matrix so, is this \Psi_j applied to a two different Dirac on the graph ? Would be good to clarify exactly what is being plotted in the legend.

seems to be the biggest limitations of the proposed approach. By not early mixing of different features one might lose the high frequencies correlations between different signals defined on a single graph.

P4. IMO capsule is not such a great name / already used in ML by Hinton's capsule etc... Why not simply 'moments' or 'statistics' ?

'We can replace (3) with normalized moments of x ... how exactly do you normalize ?

---

> ### Author Response · Authors · 2018-11-26
> **Thank you for the detailed and helpful comments! Let us try to respond point by point.**
>
> Thank you for the detailed and helpful comments! Let us try to respond point by point.
>
> Regarding the lack of theoretical results, as we mention in the paper, we focus here on practical and applicable aspects of geometric scattering as an extension and generalization of the Euclidean scattering transform. In particular, we aim to establish the capacity of our proposed scattering features to provide a rich representation of graph data. We note that even in the classical Euclidean case, theoretical results are mostly available only for stability properties (e.g., Lipshitz stability to groups of deformations), while the capacity of the scattering transform to capture rich representations of signals is established via a variety of applications and numerical experiments. Therefore, we follow the same approach here. To better clarify our focus, we added Section 3.2 that discusses these stability and capacity aspects, while positioning our work as complementary to [Zou and Lerman, 2018] and [Gama, et al, 2018] that focused on theoretical results (indeed, both prove very nice stability results, which we expect to also reflect on our construction), but lack thorough numerical experiments. Therefore, we regard a strong part of our contribution as complementary to them, as here we both fill in missing numerical validation and potentially open up a new path for theoretical investigation into the capacity of geometric (or graph) scattering, especially with new results added in the revised version of the paper demonstrating the application of geometric scatting to data exploration.
>
> Regarding parameter choices, the scattering transform essentially has two configurable parameters: the maximum scale J, which is chosen automatically here based on the diameter of the graphs, and the number of moments Q, which was chosen based on cross validation tuning, per standard practices in tuning hyperparameters in supervised learning tasks. We note that other hyperparameters that relate to the neural network classifier were also chosen in a similar manner, but are not the focus of this paper. Further, in the revised version we added classification with RBF kernel SVM, which achieves state of the art results on social network data (in particular, REDDIT B and REDDIT 5K, but also in aggregate). Therefore, our revised results reduces the dependence of presented results on hyperparameter tuning, as it does not rely solely on multiple fully connected layers (with associated tuning challenges) to establish the performance of our method on graph classification.
>
> Regarding training methodology, we believe there may be a misunderstanding here. As stated and explained in Appendix E (in the revised version), we use standard 10-fold cross validation for classification experiments, following the standard practice in other works on graph classification, which allows for reliable comparison of our results to them. As part of this procedure, each fold in fact trains the classifier on 90% (not 10%) of the data, and tests on the remaining 10%.
>
> Regarding the p2 comment: we agree that some correlations may theoretically be lost by not mixing features prior to the application of geometric scattering, and perhaps this might be an interesting idea for future work (e.g., in conjunction with random projections or other mixing strategies). However, here we focus on exploring the properties of the scattering transform itself, with emphasis on its capacity to provide rich representations of graphs, even without special preprocessing such as mixing features. Indeed, with the addition of new results, both in classification (e.g., SVM on scattering features obtaining state of the art results on social networks data) and data exploration (revealing relations between enzyme classes), we believe we provide sufficient indication to the viability of geometric scattering even without early feature mixing.
>
> Regarding the p4 comments: (1) We revised the caption of Fig. 1 to explain we indeed apply the filter matrices to Diracs on the graph. (2) We agree with the terminology suggestion and revised to use “moments” rather than capsule. (3) When we mention normalized moments, we mean in the standard statistical sense (also referred to sometimes as “standardized moments”), which are also detailed immediately after (mean, variance, skew, and kurtosis). We clarified in the revision that these are “ normalized (i.e., standardized) moments “

---

### Official Review · AnonReviewer2 · 2018-11-02
**Interesting construction but limited novelty**

**Rating:** 5
**Confidence:** 4

**Review:**

The authors propose an advance in geometric deep learning based on a geometric scattering transform using graph wavelets defined in terms of ran- dom walks on the graph. The paper is well written, easy to understand also for a not-so-tech audience but nevertheless precise in all the mathematical details.
Intro and references are satisfactory, and also the experimental section is sufficiently convincing. However, there are two big issues undermining the overall structure of the manuscript:
a) the theoretical novelty w.r.t. (Zou & Lerman, 2018) and (Game, 2018) is partial and rather technical, so the originality of the present manuscript is limited
b) the improvement w.r.t. to other published method is rather small, so the performance gain is only partially justified by the quite complex theoretical construction.

---

> ### Author Response · Authors · 2018-11-26
> **Thank you for the helpful comments! Let us respond first to (b) and then (a).**
>
> Thank you for the helpful comments! Let us respond first to (b) here, and then to (a) in a subsequent post.
>
> For (b), we have augmented the numerical experiments in the submitted version of the paper with the following additional experiments.
>
> In the revised Section 4.1, rather than using fully connected layers at the backend, we used an SVM classifier with RBF kernel (hyperparameters chosen via cross validation), which improved some of the classification results a small, but potentially significant, amount. In particular, the geometric scattering followed by SVM achieved state of the art results on two data sets, REDDIT B and REDDIT 5K, compared to the other methods presented in the paper. Further, we now also achieve state of the art results in aggregate on social network data.
>
> We have also investigated geometric scattering in the context of data exploration of biochemistry data, added in the new Section 4.2 :
>
> -  First, we examined the intrinsic dimension of the geometric scattering coefficients by using a simple PCA explained variance test, i.e., by counting the number of principal components needed to capture 90% of the variance. We see that, according to this measure of dimensionality, the geometric scattering coefficients give intrinsically low dimensional descriptions of the data. It is likely that other graph CNNs do the same, but we emphasize that the graph convolution filters in these networks are task dependent, whereas the geometric scattering wavelet filters are task independent and more suitable for data exploration.
>
> -  Next, using only the PCA projections from the previous bullet (capturing 90% of the variance of the geometric scattering coefficients), we carried out the SVM classification and found the classification rate does not drastically decrease. In particular, the geometric scattering coefficients aggregate useful information into the primary dimensions of variability, and indeed these dimensions describe the majority of class differences.
>
> -  We have added new experiment indicating the degree to which to the geometric scattering coefficients separate classes within the datasets, which we hypothesize are the result of systematic structural differences in the graphs, and captures relations between them. We focused here on the ENZYMES dataset, where we show that this hypothesis is at least partially validated, and the geometric scattering coefficients are able to reasonably separate the classes into lower dimensional subspaces of the full feature space, even without any training via supervised learning.
>
> -  Moreover, we show that the relations between subspaces of enzyme classes (labeled by enzyme commission numbers) here can in fact be used to reveal exchange preferences between classes during enzyme evolution, which we validate by comparing to established knowledge from [Cuesta et al., 2015].
>
> We believe these new results provide a significant improvement, and indicate the added value and contribution of the proposed geometric scattering in graph data analysis. We also remark that the construction itself is not that complicated (at least compared to other geometric deep learning methods). The algorithm itself is merely a graph convolution with multiscale graph wavelet filters, followed by an absolute value nonlinearity, which can be repeated. We emphasize there is no training of these filters, so the training time is completely determined the time needed to train the fully connected layers or the SVM, which is not much. It additionally eliminates the need to pay considerable attention to the computational complexity of the graph convolution, since it does not need to be repeated numerous times in the training process, but rather is a one time computational cost. Of course for very large graphs, though, certain considerations should still be taken into account, but this is true for other methods as well.

---

> > ### Author Response · Authors · 2018-11-26
> > **Part II of our response - now addressing (a)**
> >
> > For (a), we agree that the construction presented here is similar to those of [Zou and Lerman, 2018] and [Gama et al, 2018], particularly the latter. We point out that while [Zou and Lerman, 2018] and [Gama et al, 2018] prove some very nice theoretical properties of their versions of graph scattering transforms, their numerical experiments do not give much indication as to whether this theory is relevant to practical graph learning tasks. In this paper we have shown that, indeed, they potentially are. Furthermore, both [Zou and Lerman, 2018] and [Game, et al, 2018] focus primarily on stability results, i.e., if two graphs are similar (e.g., of the same class), will the resulting scattering coefficients also be similar? Under certain theoretical frameworks, they prove positive statements along these lines. Some of the new experiments we have added in, though, aim at shedding light on the converse to this question, namely: if two graphs are dissimilar (e.g., in different classes), are the geometric scattering coefficients sufficiently different to separate the classes? While we don’t prove any theoretical results along these lines, the new numerical experiments added into the revised version of the paper are sufficiently positive that they open up this question for further numerical and theoretical work. A discussion along these lines was added as the new Section 3.2 to differentiate these stability vs capacity properties and distinguish our focus from the one in these two related works.
> >
> > We remark that even in the classical case, theoretical results on scattering transforms are mostly available only for stability properties, while the capacity of the scattering transform to capture rich representations of signals is established via a variety of applications and numerical experiments. We would thus view this paper as complementary to the works of [Zou and Lerman, 2018] and [Gama, et al, 2018], as it both fills in missing numerical validation and potentially opens up a new path for theoretical investigation.
> >
> > We hope that with these additions we were able to sufficiently improve the quality of the presented results to warrant an update to the reviewers score.

---

### Meta-Review · Area_Chair1 · 2018-12-15
**Lack of theoretical novelty voiced as one of main issues.**

**Confidence:** 5
**Recommendation:** Reject

**Metareview:**

AR1 is concerned about the overlap of this paper with Gama et al., 2018 as well as lack of theoretical analysis and poor results on REDDIT-5k and REDDIT-5B datasets. AR2 reflects the same concerns (lack of clear cut novelty over Zou & Lerman, 2018, Game, 2018. AR3 also points the same issue re. lack of theoretical results. The austhors admit that Zou and Lerman, 2018, and Gama, 2018, focus on stability results while this submission offers empirical evaluations.

Unfortunately, reviewers did not find these arguments convincing. Thus, at this point, the paper cannot be accepted for publication in ICLR. AC strongly encourages authors to develop their theoretical 'edge' over this crowded market of GCNs and scattering approaches.